# Characteristics of Interpolyelectrolyte Complexes Based on Different Types of Pectin with Eudragit^®^ EPO as Novel Carriers for Colon-Specific Drug Delivery

**DOI:** 10.3390/ijms242417622

**Published:** 2023-12-18

**Authors:** Shamil F. Nasibullin, Julia V. Dunaeva, Lilija A. Akramova, Venera R. Timergalieva, Rouslan I. Moustafine

**Affiliations:** Institute of Pharmacy, Kazan State Medical University, 16 Fatykh Amirkhan Street, 420126 Kazan, Russia; shamil.nasibullin@kazangmu.ru (S.F.N.); venera.timergalieva@kazangmu.ru (V.R.T.)

**Keywords:** Eudragit^®^ EPO, citrus pectin, apple pectin, diclofenac sodium, interpolyelectrolyte complex

## Abstract

Given that pectin is a well-known substance used for drug delivery, we aimed to obtain and further examine the efficacy of interpolyelectrolyte complexes based on citrus or apple pectin and the Eudragit^®^ EPO for using these carriers in oral drug delivery. To characterize the physicochemical properties of these compounds, turbidity, gravimetry, viscosity, elementary analysis, FTIR spectroscopy, and DSC analysis were utilized. Diffusion transport characteristics were evaluated to assess the swelling ability of the matrices and the release of diclofenac sodium. To examine the release parameters, mathematical modeling was performed by using the Korsmayer–Peppas and Logistic equations as well. During the turbidity study, stoichiometry compositions were selected for the developed IPECs EPO/PecA and EPO/PecC at pH values = 4.0, 5.0, 6.0, and 7.0. The FTIR spectra of the complexes were characterized by an increase in the intensity of the bands at 1610 cm^−1^ and 1400 cm^−1^. According to the DSC analysis, IPEC has a certain Tg = 57.3 °C. The highest release rates were obtained for IPEC EPO/PecC_1 and EPO/PecC_4. The mechanism of drug transport from the matrices IPEC EPO/PecC, IPEC EPO/PecA_3, and EPO/PecA_4 can be characterized as Super Case II. Anomalous release (non-Fickian release) is typical for IPEC EPO/PecA_1 and EPO/PecA_2. Thus, the resulting systems can be further used for the effective delivery of the drugs to the colon.

## 1. Introduction

One of the promising ways to improve the safety of drug carriers is the usage of biopolymers. Biopolymers of a polysaccharide nature, such as pectin, sodium alginate, starch, chitosan, and inulin, are the most widespread [1,2,3]. Many studies have been conducted on the usage of natural polysaccharides in drug delivery systems. All currently used polysaccharides are described in detail in the review [4]. The systematization of polysaccharides and methods of therapeutic agent inclusion are given, and the mechanisms of release from these systems are considered. Recently, interesting results were published regarding the films with sustained diclofenac sodium release based on hallosite nanotubes and natural funori polysaccharides [5]. In one of the studies, biohybrid tablets were obtained based on hallosite and chitosan nanotubes containing sodium diclofenac and coated with alginate layers on both sides. The authors noted that the sandwich like structure of the hybrid tablets allowed to control drug release under variable physiological pH levels, mimicking the gastrointestinal tract (GIT) conditions [6].

The present work focused on the polycomplex systems based on pectin (polyanion) and a pharmaceutically acceptable basic butylated methacrylate terpolymer—Eudragit^®^ EPO (polycation). It is necessary to mention previously conducted studies involving these polymers.

Pectin is one of the most widespread and available polysaccharides in the plant world. In pharmaceutical technology, it is used as a binder [7,8,9,10]. It should be noted that pectin belongs to the category of biodegradable polymers. It is stable in the upper GIT, but it is degraded by the microflora of the colon, mainly anaerobic bacteria such as *Bifidobacterium*, *Bacteroides*, and lactobacilli of the genus *Lactobacillus* [11].

Moreover, there are many well-known developed oral DDSs, containing pectin or calcium pectinate, used in different formulations, such as tablets, particles, microparticles, pellets, and beads, which have been well discussed and systematized in detail in a review of pectin-based DDSs used for the treatment of colon cancer [12].

Biodegradable gels based on pectin and chitosan were obtained by a group under Khutoryanskiy; the rheological properties, swelling ability of gels, degradation under the action of hyaluronidase as an enzyme, and release of cisplatin were studied as well [13]. It has been shown that the use of pectins with different molecular weights makes it possible to prepare hydrogels with different rheological characteristics and swelling kinetics. It was established that the viscosity of citrus pectin hydrogel is about four times higher than for apple pectin hydrogel, which is likely associated with differences in their molecular weights (70 and 30 kDa, respectively). It has been shown that in the presence of hyaluronidase, the hydrogel undergoes degradation. Unfortunately, the authors did not evaluate the release kinetics in the presence of enzymes. The cisplatin release is regulated to a certain extent by the choice of hydrogel composition via interaction between chitosan and pectin within hydrogels.

Films based on pectin or calcium pectinate with cellulose (Aquacoat^®^ ESD30 (Elgin, IL, USA), Surelase^®^ (Colorcon, Chalfont, PA, USA) or methacrylate (Eudragit^®^ NE30D, RS30D (Evonik Industries AG, Darmstadt, Germany) polymer dispersions were obtained and studied by the research group of Semde et al. [14]. The leaching of pectin from the obtained films was studied, and a slowdown in the dissolution of pectin from films containing Eudragit^®^ RS30D was noted, which was explained by the interaction of ionized carboxyl groups of pectin with quarter amino groups of Eudragit^®^ RS30D.

The same team of co-authors studied the effect of pectinolytic enzymes on the release of theophylline from dispersions containing pectin or calcium pectinate coated with the above-mentioned cellulose or methacrylate dispersions [15]. The release of theophylline was lower in the presence of pectinolytic enzymes. The authors attribute this to the ability of pectin to hydrate, swell, and form channels through which the hydrophilic drug molecules easily diffuse into the environment. In the presence of oppositely charged polymer (Eudragit^®^ RS30D), the hydration of pectin reasonably decreases, followed by a decrease in the drug release level in the presence of pectinolytic enzymes.

Films based on pectin, HPMC, and chitosan were developed using the solvent cast method using 0.1 M hydrochloric acid or 0.1 M acetic acid in the work of other researchers (Ofori-Kwakye and Fell) [16]. The leaching of pectin in a medium mimicking the upper GIT was studied in the presence and absence of pectinolytic enzymes. The addition of enzymes led to an increase in the pectin dissolution, which they explained as the degradation of polysaccharides from the obtained films, in contrast to the previous work of the other authors (Semde et al.) [14,15]. Attention is also drawn to the polycomplex formation involving the carboxylate groups of pectin and the ammonium groups of chitosan, which influences the resulting swelling and releasing behavior of the developed DDS.

Methacrylate copolymers are widely used as bases for matrices for oral DDSs [17]. Among them, a special place is given to the polymers under the well-known trade name Eudragit^®^, produced by the Evonik Industries AG company (Darmstadt, Germany). These excipients are used in the manufacturing of tablets, granules, and micro- and nanosized particles as coatings, or as binders in the granulation stage to prepare matrix tablets. They are offered in the form of powders, granules, and 30% aqueous dispersions of methacrylate copolymers. Depending on the ratio of carboxyl and ether groups, these copolymers are dissolved at different pH values and can be different in terms of dissolution rate. They are used to obtain tablet coating materials that allow to control the release of the drug in the desirable part of the GIT. For example, Eudragit^®^ type E is a weak base terpolymer used to develop coatings that are soluble in the stomach region [17]. Many studies have been also published focused on the development of DDSs based on polymethacrylates based on their oppositely charged structures by using combinations of polycation and polyanion types [18,19,20,21,22].

Taking into account the ability of the studied polymers to interact due to the polyanion–polycation structure, the simplest way to prepare physically cross-linked systems is to mix solutions of oppositely charged polyelectrolytes (PEs), resulting in the formation of interpolyelectrolyte complexes (IPECs) in order to modify the properties of individual polymers [23,24,25,26,27,28,29,30].

A lot of findings were published using IPECs that could be obtained through the interaction of oppositely charged pharma polymers. So, IPECs involving pectin, chitosan and Eudragit^®^ RS were prepared by blending polymer solutions following film preparation by using the solvent casting method [31]. In addition, numerous studies have been conducted on the interaction of pectin and chitosan as suitable materials for the design of drug delivery systems (DDSs) [32,33,34,35,36,37,38]. Influenced by IPEC formation, polyionic hydrogels were designed as floating drug carriers [39].

Thus, the research group of Moustafine et al. studied IPECs based on Eudragit^®^ EPO (EPO) and alginate sodium [40,41]. In this research, both physicochemical and swelling-ability properties, as well as the processes, occurring within the swelling of matrices in the GIT-mimicking environments were studied. The drug releasing experiments showed that the developed DDSs are suitable for the colon-specific delivery of diclofenac sodium [42]. It is interesting to note that the usage of the same pair of oppositely charged polymers was also suitable for the encapsulation of fluorouracil or indomethacin. Developed microcapsules were firstly shown to be gastroresistant, followed by the intestinal release of included drugs [19,20].

The interaction of high- and low-viscosity alginates with EPO and the comparative evaluation of the obtained IPECs with physical mixtures of the same compositions were also studied [43]. The interaction between these polymers and the slowing down of the release of diltiazem hydrochloride from tablets was proven.

Another group of scientists conducted studies on the interaction of sodium alginate with quaternary polymethacrylates and noted a slowdown in the release of the propranolol hydrochloride from obtained gel bids [44].

In some papers, authors announced the results of studies regarding the interaction phenomena involving sodium carboxymethylcellulose and EPO macromolecules for preparing IPECs and oral levodopa delivery systems based on them [45,46,47]. Similar studies included research on IPECs based on biodegradable polymer starch and kappa carreginan [48,49]. IPECs involving naturally sulfated polysaccharides of the seaweed *Polysiphonia nigrescens* and cationized agaroses and Eudragit type E were prepared, characterized, and explored for controlled drug release in other studies [50,51]. 

IPECs obtained on the basis of pectin and EPO synthetic copolymer open up the possibility of developing a drug delivery system for colon targeting [52,53]. So, EPO and pectin are oppositely charged PEs. Both of these polymers were well studied in the above-mentioned works in various combinations. Thus, one of the possible ways to modify their structures is to include them in an IPEC structure. 

The objective of this study was to investigate the fundamental physicochemical characteristics of new IPECs prepared from different types of pectin and Eudragit^®^ EPO in aqueous solutions in order to monitor the possible structural transformation of, and composition changes in, polycomplex matrices during swelling and drug release in mimicking GIT conditions with respect to their potential application in colon-specific drug delivery. Diclofenac sodium (DS) was used as a model drug.

## 2. Results

### 2.1. Turbidity Measurements

First, the interactions between diluted solutions of pectins and EPO at different ratios of polymers were studied. When the solutions were mixed, a substantial increase in the solution turbidity measured at 600 nm was observed. The dependence of the degree of turbidity on the composition of the reaction medium is shown in Figure 1a–h.

Based on the results of turbidimetry, graphs representing a typical turbidimetric titration curve were constructed, which had a maximum at certain ratios of polymers; when an excess amount of pectin or excess quantity of EPO macromolecules was added, turbidimetry decreased and IPECs precipitated. During turbidimetric titration, interaction between two PEs occurs, which leads to an increase in turbidity in the solution as the concentration of polymers increases to a certain maximum. In the study, the addition of an excess amount of EPO or pectin again led to a drop in the turbidity value due to the segregation of the formed IPEC. The observed binding molar ratio at the maximum turbidity value corresponds to the stoichiometry of the obtained product. The further addition of PEs leads to a decrease in turbidity due to the precipitation of sediment [23,24,25,41].

### 2.2. Apparent Viscosity Measurements and Gravimetry

In the frame of our work, we synthesized IPECs with different compositions at the different pH values of the used medium. The minimum viscosity value and the maximum yields of IPEC precipitates at these points indicate that the interpolyelectrolyte reaction was completed the most; these points correspond to stoichiometry compositions. On the figures with histograms of viscosity and gravimetry for EPO—Pec, the following relationships can be noted as stoichiometry compositions: for pH = 4.0 1:1.5 for both types of pectins; for pH = 5.0 1:1 for PecC and 1:1.5 for PecA; for pH = 6.0 1:1 for both types of pectin; and for pH = 7.0 1.5:1 for PecC and 4:1 for PecA (Figure 2a–h).

### 2.3. Elemental Analyses

In order to confirm the interaction or binding ratio of each component in the solid IPEC, elementary analysis was performed. It can be assumed that as the degree of esterification of pectin grows higher, the less reactive it is and the larger amount of it is required to neutralize the oppositely charged polymer from the results of elemental analysis (Table 1). In our case, apple pectin has a high degree of esterification (73.0 ± 1.1).

### 2.4. FTIR Spectroscopy

The FTIR spectrum of the EPO/PecC complex shown in Figure 3 is characterized by an increase in the intensity of the bands at 1610 cm^−1^ and 1400 cm^−1^ in comparison with the FTIR spectrum of the physical mixture (Figure 3), while in the FTIR spectrum of the EPO/PecA complex (Figure 4), a band appears at 1610 cm^−1^. 

### 2.5. Thermal Analysis

DSC analysis was carried out on samples of the IPEC, a physical mixture of the same composition, and individual polymers used to obtain this IPEC (Figure 5). DSC thermograms show the presence of a glass transition temperature for all samples except those of citrus pectin. That is, pectin does not vitrify. The remaining samples have not only one but different glass transition temperatures (Tg), which shows that the preparation of IPECs leads to the formation of miscible polycomplex systems.

### 2.6. Determination of the Degree of Swelling of Matrices

Taking into account the pharmaceutical focus of our research, the next stage of our work was to study the diffusion transport properties of the resulting polycomplexes in comparison with physical mixtures of a similar composition and individual polymers.

One of the most important characteristics of polymers is their ability to swell, which determines their physicochemical properties and the feasibility of using them as polymer carriers of drugs.

In our work, we studied the swelling kinetics of IPECs obtained at the identified EPO/Pec ratios in media mimicking the GIT conditions (7 h) in comparison with individual polymers and physical mixtures of a composition similar to the composition of IPEC EPO/Pec.

Figure 6a,b show the swelling curves of polycomplexes (symbols are presented in Table 2); Figure 6c,d show a physical mixture and an individual polymer (pectin). There is no swelling profile for Eudragit^®^ EPO. This is due to the fact that this copolymer, which has a basic character, dissolves in an acidic environment. Therefore, tablets prepared from this copolymer dissolved in the medium mimicking the stomach fluid (pH = 1.2).

### 2.7. In Vitro Drug Release Test

An oral system based on polycomplexes is exposed to environments with different pH values (from 1.2 to 7.5), which affects the rate of drug release in various parts of the GIT; therefore, we determined the rate of DS release from polymer matrices based on IPECs while mimicking the GIT conditions.

Figure 7 shows the release profiles of DS from matrices based on IPEC EPO/Pec. The release of DS from tablets can be characterized as being of the delayed type. For a more detailed analysis of the transport mechanisms of DS from IPEC matrices, mathematical modeling of the drug release processes was carried out according to the Korsmeyer–Peppas equation [54] (Table 3).

However, the fitting parameters, and especially the correlation factor, did not fully describe the release curve; thus, we used another function—a Logistic function. A Logistic function describes the exponential rise in concentration and its levelling off as the experimental time increases. This function was more suitable for our system and had better correlation coefficients.

## 3. Discussion

The Eudragit EPO polymer has been used in the pharmaceutical industry for many decades in oral dosage forms and, therefore, it can be concluded that this polymer is safe [55]. Pectin is a polysaccharide of natural origin, obtained from apples and citrus fruits, and it is biodegradable as it is digested in the intestines. Thus, IPECs obtained on the basis of studied polymers are safe for use as delivery systems.

Turbidimetry was performed to assess the interaction between polymers. The presented figures show typical curves of the turbidimetric titration of EPO and pectin solutions and the converse (pectin–EPO) in a medium of estimate pH values: 4.0, 5.0, 6.0, and 7.0.

It should be noted that at pH = 4, turbidity maxima were observed at a ratio of EPO/Pec polymers of 4:6 (corresponds to a 1:1.5 molar ratio), regardless of the mixing order during synthesis, both for apple and citrus pectins (Figure 1a,b). At pH = 5, in the case of PecC, the maximum was observed at a polymer ratio of 5:5 (corresponds to the equimolar ratio), and for PecA, 4:6 (Figure 1c,d). At pH = 6, the maxima were, in both cases, at a polymer ratio of 5:5 (Figure 1e,f). And, at pH = 7, the polymer ratio of 6:4 (corresponds to a 1.5:1 molar ratio) was typical for PecC, and it was 8:2 (corresponds to a 4:1 molar ratio) for PecA (Figure 1g,h). Therefore, most of the turbidity of the system corresponded to maximum interaction between copolymers.

The results of the apparent viscosity and gravimetric analysis from the copolymers combinations are shown in Figure 2a–h. The decrease in the apparent viscosity values of the supernatant of EPO–pectin mixture solutions was observed in the system; it was shown that the IPEC was formed in the investigated medium and was removed through centrifugation. Therefore, using turbidimetry, apparent viscosity, and gravimetry for the estimation of stoichiometric ratios (EPO/pectin) leads to receiving the concurring results. The fraction of polycation (EPO) incorporated in the polycomplex increases as the pH rises. Meanwhile, the first, synthesized at pH 4.0 (denoted as IPEC EPO/PecC_1 or IPEC EPO/PecA_1), has a composition with 1.5 excess of pectin. The IPECs that were prepared at pH 5.0 and 6.0 had compositions close to the equimolar composition (denoted as IPEC EPO/PecC_2, IPEC EPO/PecC_3, IPEC EPO/PecA_2, and IPEC EPO/PecA_3); IPEC EPO/PecC_4 or IPEC EPO/PecA_4 (synthesized at pH 7.0) contained a 1.5-fold excess of EPO. Thus, the polycomplexes were enriched with the less ionized component (charge density on EPO chains > 0 at pH = 7.0). On the other hand, the incorporation of the polyanion (pectins) decreased due to the progressive increase in the fraction of ionized carboxylic acids. This also increased its reactivity. These phenomena were in agreement with those observed in our previous studies with Eudragit^®^ E/Eudragit^®^ L100 (L100-55), Eudragit^®^ E/alginate, Eudragit^®^ E/kappa-carrageenan, or Eudragit^®^ E/CMC systems and with reports from the literature [41,42,45,48,55,56].

To assess the interaction between PEs, FTIR spectra of IPEC samples (Figure 3 and Figure 4 and physical mixtures (Figure 3 and Figure 4 of the same composition were recorded. The FTIR spectra of the IPECs were characterized by an increase in the intensity of the bands at 1610 cm^−1^ and 1400 cm^−1^, which could be assigned to the absorption band of the carboxylate groups that formed the ionic bonds with the protonated dimethylamino groups of EPO. Furthermore, the presence in the complexes of the band at 2450 cm^−1^ corresponded to the absorption of ionized dimethylamino groups of EPO in relation to the polycomplex with the carboxylate groups of pectin [41,42,57,58]. This band was absent in the FTIR spectra of the physical mixture. Therefore, prepared polycomplexes were stabilized by macromolecular ionic bonds according to the presented scheme (Figure 8).

According to the elemental analysis (Table 1), the ratios obtained from the analysis were very close to the molar ratios at which the samples were synthesized. But, it should be noted that in comparison with citrus pectin IPECs, a slightly larger number of apple pectin macromolecules was required due to their higher esterification values (73.0 ± 1.1).

Thus, the results of physicochemical characterization confirmed the formation of IPECs between oppositely charged PEs at pH values from 4.0 to 7.0. The structure of synthesized Eudragit EPO/pectin IPECs, due to differences in the starting change density of the interacting macromolecules, depends on the molar ratio of each component in the polyion mixture and correlates with their estimated stoichiometric compositions, showing a change from 1:1.5 to 1.5:1. 

According to the DSC analysis (Figure 5), mDSC thermograms were obtained with different Tg temperatures: for EPO, Tg = 50.1 °C; for the physical mixture EPO/PecC, Tg = 53.4 °C; and for the IPEC, Tg = 57.3 °C. Thus, with the involvement of pectin in polycomplex structures, the resulting Tg of the included IPEC EPO increased. It is interesting to note that in the thermogram of the physical mixture, we do not see two glass transition temperatures despite the fact that this is a mechanical mixture of two polymers, pectin and EPO, most likely due to the fact that pectin itself does not vitrify. However, a difference in Tg between the physical mixture and IPEC is observed due to the fact that the pectin macromolecules in the IPEC structure are ionically bounded to EPO sequences, unlike the physical mixture, which proves that the IPEC was successfully prepared (the single Tg in IPEC was estimated at 57.3 °C, and it was a higher Tg value than that of the individual EPO, which had a Tg of 50.1 °C).

The next stage of research was an assessment of diffusion transport properties, namely the kinetics of swelling of polymer matrices, while mimicking the GIT conditions. We used a sequential change in pH during the study of swelling kinetics and the drug delivery experiment because we wanted to simulate drug molecule transition through the gastrointestinal tract. Our systems included polymers that were pH- and time- dependent. The IPEC matrices swelled when passing through the GIT; thus, drug delivery occurred because the matrices swelled as the pH changed and as the matrix residence time in the environments increased. According to the results, IPEC samples obtained at pH = 4.0 in a ratio of 1:1.5 based on citrus pectin (Figure 6a) and at pH = 4.0 and pH = 5.0 in a ratio of 1:1.5 based on apple pectin (Figure 6b) had been disintegrated: by the sixth hour (pH = 7.4) based on PecC and within only two hours (pH = 5.8) based on PecA. This was possibly due to the composition of these IPECs, which had contained excess amounts of pectin in a polycomplex structure. Since pectin is a hydrophilic polymer, it is easily hydrated in aqueous salt medium and the tablet matrix is easily disintegrated. A similar result was described in theophylline release studies [24]. Perhaps apple pectin is more hydrophilic, since the IPEC based on it disintegrated faster—after 2–2.5 h at pH = 5.8 (Figure 6b). In case of the physical mixture, the dimethylamino groups of Eudragit^®^ EPO and the carboxyl group of pectin are in free form; therefore they were easily ionized in the analyzed medium (pH = 5.8), contributing to the swelling of the hydrated matrix (Figure 6c,d). The subsequent decrease in swelling rates (at pH = 6.8) happened due to the gradual dissolution of the ionized polysaccharides. At pH = 7.4, a slowdown in the dissolution of the physical mixture matrices was observed and compared to pectin, and it had apparently happened due to the formation of ionic bonds between two oppositely charged polymers within matrices. Concurring results were observed in our previously published studies on oppositely charged systems made up of physical mixtures of chitosan–L100-55 (L100) and EPO–alginate sodium [42,59]. The profile of the pectin matrix had the same character, but due to the intensity of the dissolution of the polymer itself at pH = 7.4, it had lower values by the end of the experiment. It is interesting to note that the swelling ability of the IPEC and physical mixtures based on apple pectin was significantly higher than that based on citrus pectin (Figure 6a–d). It should be concluded that all IPEC samples are suitable for further evaluation as carriers for oral DDSs.

According to drug delivery results, DS showed ‘intestinal’-type release profiles (Figure 7a,b). The IPECs in this study belong to the category of pH- and time-dependent colon-specific DDSs because the release rate is minimal at a lag-time period of time, followed by comparatively rapid drug release rate at a site corresponding to the colon region [52,53,59]. The swelling ability of matrixes that were prepared from IPECs can be tuned as per their composition. This gives the possibility to tune the ratio of hydrophilic and hydrophobic sequences in the structures of IPECs. Similar studies based on sodium alginate and Eudragit^®^ E have been reported by our research group previously [42].

The highest release rates were shown by IPECs based on citrus pectin IPEC_EPO/PecC_1 and IPEC_EPO/PecC_4 (Figure 7a), which was also consistent with the swelling properties of these matrices, although the matrix based on IPEC_EPO/PecC_1 disintegrated at pH = 7.4, which may have been due to the fact that this sample was obtained at a more acidic pH value (pH = 4.0). The mechanism of drug transport from the matrix can be characterized as Super Case II since the release exponential (*n*) is greater than 1 (Table 3).

If we compare the data on the swelling and release of IPEC on PecA (Figure 7b), we can note that matrices based on samples IPEC_EPO/PecA_1 and IPEC_EPO/PecA_2 swell and collapse in a slightly acidic environment (pH = 5.8), and when assessing the release, they show a low level of DS release. The matrices made up of IPEC_EPO/PecA_3 and IPEC_EPO/PecA_4 swell well, but the last one has more swelling-degree values when the matrix has transferred to pH = 7.4, and also surpasses other samples based on apple pectin in terms of release level. This IPEC sample was obtained at pH = 7.0, which was very close to the pH value of the release medium (pH = 7.4). Probably, this was the reason why the swelling and release values were higher than those from the other samples. According to the mathematical calculation (Table 3), IPEC_EPO/PecA_3 and IPEC_EPO/PecA_4 have a release exponent that is greater than 1 (*n* > 1), so the mechanism of drug transport from the matrix is Super Case II. This means that the drug transport mechanism was associated with stresses and state transition in hydrophilic polymers that swell in GIT-mimicking fluids. Anomalous release (non-Fickian release) is typical for IPEC_EPO/PecA_1 and IPEC_EPO/PecA_2 (0.5 < *n* < 1) [54].

The Korsmeyer–Peppas equation is used to describe drug release from a polymeric system considering non-Fickian mechanisms. Thus, we used the Korsmeyer–Peppas model, because the systems were obtained on the basis of polymers; however, this model did not take into account the induction time. For more detailed fitting, another model was chosen. One example of the uses of a Logistic model was to describe the release from functionally graded materials, various composite media in which constitutive property elements change smoothly and continuously from one surface to another [60].

The parameters of the logistic model equation show the following:
y = *D*(*x*)—drug transport according to the diffusion equation;A1 = *D*(max)—maximum diffusion;A2 = *D*(min)—minimum diffusion;p = −λ·(x − σ)—the product of the inversely proportional width of the path of the drug (λ) from the tablet and the difference between the release value (x) and the location of the center of diffusion of the drug (σ) from the tablet;x0—release value at maximum diffusion.

When comparing systems based on apple and citrus pectin, the p value is greater on the basis of apple pectin; that means that the drug molecules travel a longer distance during diffusion. A2 diffusion values are greater in systems with citrus pectin. But, the values of x0 and *R*^2^ are comparable.

A logistic function is one describing the exponential rise in concentration and its leveling off as the experimental time increases. This function is more suitable for our system and shows better correlation coefficients. Thus, release in this model occurs due to the diffusion of the drug from the tablet while, in the Korsmeyer–Peppas model, it occurs due to the swelling of the polymer matrix and relaxation of the polymer chains.

Thus, the resulting systems are potentially suitable for colon-specific drug delivery; they show a characteristic lag phase in the first hours of release in the stomach- and small intestine-mimicking fluids, followed by an increase in the quantity of DS released after moving to the large-intestine-mimicking conditions with pH values of 6.8 and 7.4.

## 4. Materials and Methods

### 4.1. Materials

Pectin from citrus peel (Poly-D-galacturonic acid methyl ester, Galacturonic acid ≥ 74.0%, PC) and pectin from apple (Poly-D-galacturonic acid methyl ester, degree of esterification 70–75%, PA) were used as polyanions (Merck group, Sigma-Aldrich, St. Louis, MO, USA). Eudragit^®^ E PO (EPO), a terpolymer of *N*,*N*-dimethylaminoethyl methacrylate (DMAEMA) with methylmethacrylate (MMA) and butylmethacrylate (BuMA) (PDMAEMA–co–MMA–co–BuMA) (molar ratio 2:1:1, MW 150 kDa), was used as a cationic copolymer (Evonik Industries AG, Darmstadt, Germany). As a model drug substance, diclofenac sodium (DS), from Merck (Sigma-Aldrich, St. Louis, MO, USA), was used.

### 4.2. Methods

#### 4.2.1. Turbidity Measurements

Polymer solutions were prepared at various ratios at a concentration of 0.0002 M. Mixing was carried out in two orders. Using a magnetic stirrer RET control visc-white (IKA® Werke GmbH, Staufen, Germany), the systems were brought into equilibrium and the degree of turbidity of the solutions was determined via the value of the optical density. Both the influence of the mixing order and the ratio of PE solutions were studied (Table 4). The turbidity of each sample solution was determined at 600 nm (a wavelength where no absorption due to the polymers occurred) using a Lambda 25 spectrophotometer (PerkinElmer, Waltham, MA, USA).

#### 4.2.2. Apparent Viscosity Measurements

Solutions of Pec and EPO in various molar ratios were prepared (Table 5). The resulting solutions were adjusted to the required pH values after dissolving of polymers (4.0; 5.0; 6.0; 7.0). The systems were brought to a state of equilibrium through prolonged stirring on a magnetic stirrer at 700 rpm within 3 min. The resulting systems were centrifuged at 5000 rpm for 30 min, then the precipitate was separated through filtration through a glass filter (POR-100), an aliquot of 10.0 mL was taken, and the solution outflow time was measured at least 3 times using a stopwatch with an accuracy of 0.1 s. The relative viscosity of the solution was determined using the following formula:$$\eta =\frac{\tau }{\tau_{0}}-1,$$

Here,

*η*—relative viscosity of the solution;*τ*—solution out flow time, s;*τ*_0_—solvent flow time, s.

#### 4.2.3. Gravimetry

Aqueous solutions of Eudragit^®^ EPO and pectin were prepared at different concentrations (Table 5) and mixed in two orders. The resulting systems were kept for 7 days, then the supernatant liquid was drained off and the precipitate was dried at room temperature for 2 days, followed by drying in a vacuum oven (vacuum oven VD 23, Binder, Tuttlingen, Germany) at a temperature of 40 °C to constant weight and being weighed on an analytical balance with an accuracy of 0.0001.

#### 4.2.4. Synthesis of Solid IPEC

Synthesis of a new IPEC between countercharged type of a cationic methacrylate terpolymer (Eudragit^®^ EPO) and anionic polysaccharide (pectin) were determined at different pH values from 4.0 to 7.0, depending on copolymer solubility. The PE solutions were mixed in different orders and in different molar ratios, from 5:1 to 1:5, to a constant final concentration of 0.0032 g/mL. Eudragit^®^ EPO was dissolved in 0.005 M acetic acid and pectin in 0.005 M NaOH. Then, the pH was adjusted to the required value (4.0; 5.0; 6.0; 7.0) by adding 0.005 M NaOH or 0.005 M acetic acid. Next, a pectin solution (50 drops/min) was gradually (dropwise) added to the Eudragit^®^ EPO solution through a separating funnel. After complete precipitation of the IPEC precipitate, the supernatant liquid was decanted, and the complex itself was repeatedly washed with distilled water. The resulting polycomplex was dried for 3–4 days at room temperature, followed by drying to constant weight at 40 °C under vacuum (vacuum oven VD 23, Binder, Tuttlingen, Germany). Then, IPEC was crushed and sifted through a sieve with a hole diameter of 0.25 mm.

#### 4.2.5. Elemental Analyses

The compositions of the dried IPC samples were investigated through elemental analysis using a CHNS/O Elemental analyzer Thermo Flash 2000 (Thermo Fisher Scientific, Paisley, UK) and calculated as Z = [EPO]/[PecC] (or [PecA]) (mol/mol). The vacuum-dried samples (at 40 °C for 2 days) were weighed into a crucible on an XP6 Excellence Plus XP micro balance (Mettler Toledo, Greifensee, Switzerland). The crucibles with samples were packed and placed into the combustion reactor via autosampler. Temperature in the oven was 900 °C, and the gas flow rate was 10 mL/min. Calibration of the instrument was performed with atropine standard (Thermo Fisher Scientific, Paisley, UK). Eager Xperience v1.3. Data Handling Software (07/2014) was used to analyze the results. Tests were performed in triplicate.

#### 4.2.6. FTIR Spectroscopy

ATR-FTIR spectra were recorded using a Nicolet iS5 FTIR spectrometer (Thermo Scientific, Waltham, MA, USA) using the iD5 smart single-bounce ZnSe ATR crystal. The spectra were analyzed using OMNIC spectra software (version 8.2.387).

#### 4.2.7. Thermal Analysis

Modulated DSC (mDSC) measurements were carried out using a Discovery DSC™ (TA Instruments, New Castle, DE, USA) equipped with a refrigerated cooling system (RCS90). TRIOS™ software (version 3.1.5.3696) was used to analyze the DSC data (TA Instruments, New Castle, DE, USA). Tzero aluminum pans (TA Instruments, New Castle, DE, USA) were used in all calorimetric studies. The empty pan was used as a reference and the mass values of the reference pan and of the sample pans were taken into account. Dry nitrogen was used as a purge gas through the DSC cell at 50 mL/min. Indium and n-octadecane standards were used to calibrate the DSC temperature scale; enthalpic response was calibrated with indium. Calibration of heat capacity was achieved using sapphire. Initially the samples were cooled from room temperature to 0 °C, then kept at 0 °C for 5 min and analyzed from 0 to 250 °C. The modulation parameters used were as follows: 2 °C/min heating rate, 40 s period, and 1 °C amplitude. Glass transition temperatures were determined using the reversing heat-flow signals. All measurements were performed in triplicate.

#### 4.2.8. Preparation of Tablets

In order to determine the degree of swelling, flat-faced 100 mg IPEC compacts with 8 mm diameter were prepared by compressing the given quantity of powders (EPO, PecC, PecA, PMs and IPECs) at 2.45 MPa using a hydraulic press (PerkinElmer, Waltham, MA, USA). For dissolution testing, flat-faced 150 mg compacts (100 mg of DS and 50 mg polymer carrier) with 8 mm diameter were prepared via powder compression at 2.45 MPa using a hydraulic press (PerkinElmer, Waltham, MA, USA)

#### 4.2.9. Determination of the Degree of Swelling of Matrices

A 0.1 M, hydrochloric acid solution (pH = 1.2 for 1 h) and phosphate buffers (pH = 5.8 for 2 h, pH = 6.8 for 2 h; pH = 7.4 for 2 h) were chosen as model media mimicking the GIT [58]. The polymeric matrix was placed in a tarred basket (from the dissolution test equipment), which was immersed into a thermostated bath (37 °C ± 0.5 °C). The volume of the swelling medium was 40 mL. The degree of swelling was determined every 30 min; the basket was removed from the medium, dried using filter paper, and weighed. The degree of swelling (H, %) was calculated as
$$H\;\left(\%\right)=\frac{m_{2}-m_{1}}{m_{1}}\times 100,$$
in which m_1_ is the weight of the dry sample and m_2_ is the weight of the swollen sample.

#### 4.2.10. In Vitro Drug Release Test

The release of DS from the matrix tablets in GIT-mimicking conditions was carried out in a dissolution tester, the DT-828 (Erweka, Langen, Germany), at 37 ± 0.5 °C, by using the USP I Apparatus (Basket Method). The basket rotation speed was 100 rpm and the volume of the medium was 900 mL. The release was investigated for 7 h under GIT-mimicking conditions, wherein the pH of the release medium was gradually increased [58]: 1 h in 0.1 M hydrochloric acid (pH = 1.2), then 2 h in phosphate buffer solution with pH = 5.8, then 2 h in phosphate buffer solution with pH = 6.8, and finally 2 h in phosphate buffer solution with pH = 7.4. Aliquots (5 mL) of solution were taken every 30 min, and the volume of medium was made constant by adding fresh dissolution medium. The amounts of DS released in the dissolution medium were determined using UV/Vis-spectrophotometry at 276 nm (Lambda 25; PerkinElmer, Waltham, MA, USA). Results are given as the mean values of three determinations ± standard deviations.

#### 4.2.11. Statistical Analysis

All experiments were carried out in triplicate. Microsoft Office Excel 2007 software was used for statistical analysis. Mean values ± standard deviations were calculated using one-way analysis of variance (ANOVA) and *t*-test (Two-Sample Assuming Equal Variances), where probability was *p* < 0.05 as a significant criterion.

## 5. Conclusions

During the turbidity study, stoichiometry points were selected for the studied pairs of polymers, EPO/PecA and EPO/PecC, at pH values = 4.0, 5.0, 6.0, and 7.0. These results were reproduced and confirmed in apparent viscosity and gravimetry studies at higher concentrations. FTIR spectroscopy and DSC analysis data proved that the IPEC had been successfully prepared. The stoichiometric ratios obtained from the elementary analysis were very close to the molar ratios at which the samples had been synthesized. The swelling ability of the IPEC and physical mixtures based on apple pectin is significantly higher than that based on citrus pectin. All test samples are suitable for further evaluation for drug release. According to drug delivery results, DS shows ‘intestinal’-type release profiles. IPECs in this study belonged to the category of pH- and time-dependent colon-specific DDSs because their release rate was minimal for an initial lag-time period, followed by a comparatively rapid drug release rate at a site of the colon region. The highest release rates were shown by IPEC EPO/PecC_1 and EPO/PecC_4, which is also consistent with the swelling properties of these matrices. The mechanism of drug transport from the matrices IPEC EPO/PecC can be characterized as Super Case II. IPEC EPO/PecA_3 and EPO/PecA_4 have a release exponent greater than 1 (*n* > 1), so the mechanism of drug transport from the matrix is Super Case II. Anomalous release (non-Fickian release) is typical for IPEC EPO/PecA_1 and EPO/PecA_2. Another function was selected as a fitting model—Logistics. A Logistic function is one describing the exponential rise in concentration and its leveling off as the experimental time increases.

In our work, new oral diclofenac sodium delivery systems were successfully prepared by using polycomplexes made from apple or citrus pectins with Eudragit^®^ type EPO. The developed systems are potentially suitable for colon-specific drug delivery since they show a characteristic lag phase in the first hours of release in stomach- and small-intestine-mimicking fluids followed by an increase in the amount of drug release after moving to the colon-mimicking conditions with pH values of 6.8 and 7.4.

## Figures and Tables

**Figure 1 ijms-24-17622-f001:**
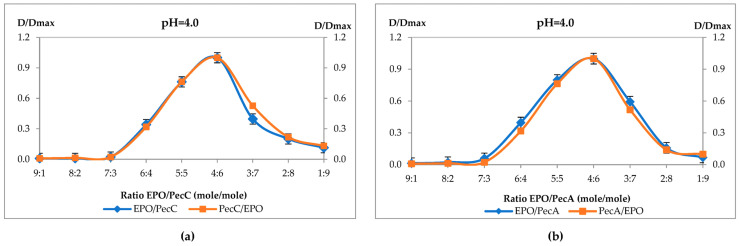
Dependence of the degree of turbidity on the composition of the reaction medium: (**a**) at pH = 4.0 EPO/PecC, PecC/EPO; (**b**) at pH = 4.0 EPO/PecA, PecA/EPO; (**c**) at pH = 5.0 EPO/PecC, PecC/EPO; (**d**) at pH = 5.0 EPO/PecA, PecA/EPO; (**e**) at pH = 6.0 EPO/PecC, PecC/EPO; (**f**) at pH = 6.0 EPO/PecA, PecA/EPO; (**g**) at pH = 7.0 EPO/PecC, PecC/EPO; and (**h**) at pH = 7.0 EPO/PecA, PecA/EPO.

**Figure 2 ijms-24-17622-f002:**
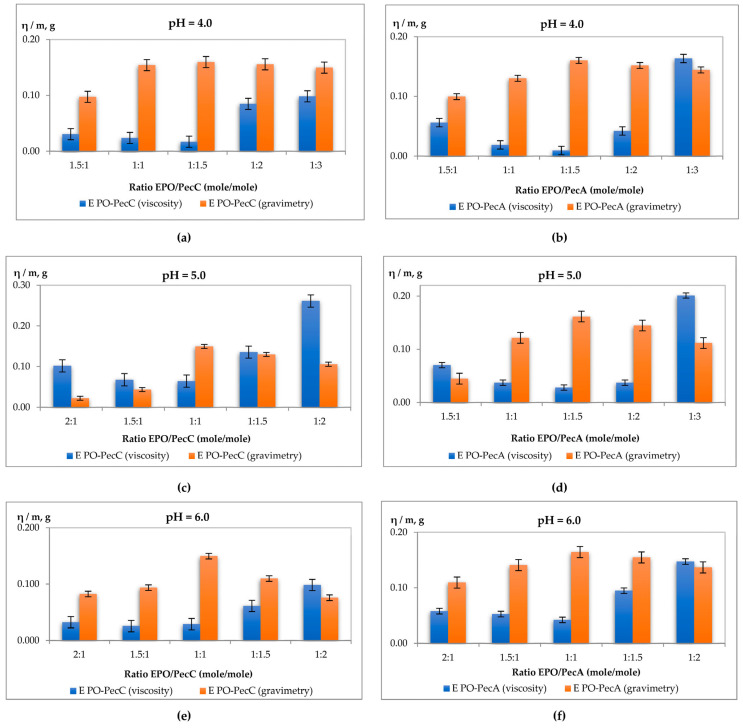
Dependence of the viscosity and gravimetry on the composition of the reaction medium: (**a**) at pH = 4.0 EPO/PecC, PecC/EPO; (**b**) at pH = 4.0 EPO/PecA, PecA/EPO; (**c**) at pH = 5.0 EPO/PecC, PecC/EPO; (**d**) at pH = 5.0 EPO/PecA, PecA/EPO; (**e**) at pH = 6.0 EPO/PecC, PecC/EPO; (**f**) at pH = 6.0 EPO/PecA, PecA/EPO; (**g**) at pH = 7.0 EPO/PecC, PecC/EPO; and (**h**) at pH = 7.0 EPO/PecA, PecA/EPO.

**Figure 3 ijms-24-17622-f003:**
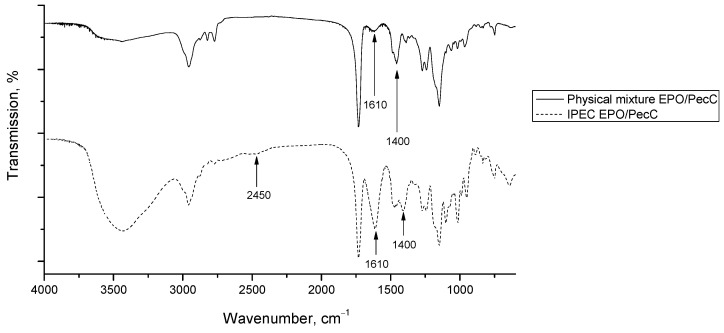
The FTIR spectrum of the physical mixture EPO/PecC and of the IPEC EPO/PecC.

**Figure 4 ijms-24-17622-f004:**
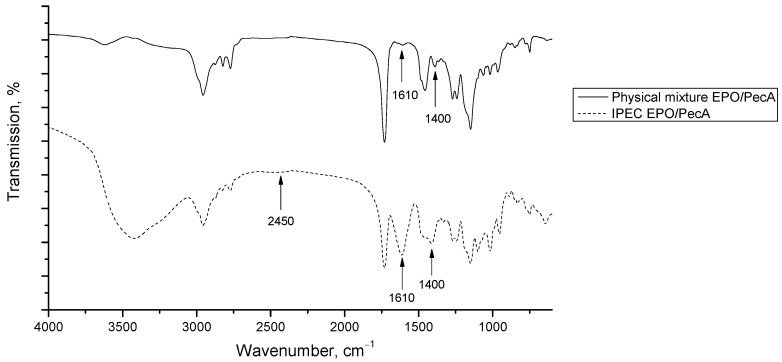
The FTIR spectrum of the physical mixture EPO/PecA and of the IPEC EPO/PecA.

**Figure 5 ijms-24-17622-f005:**
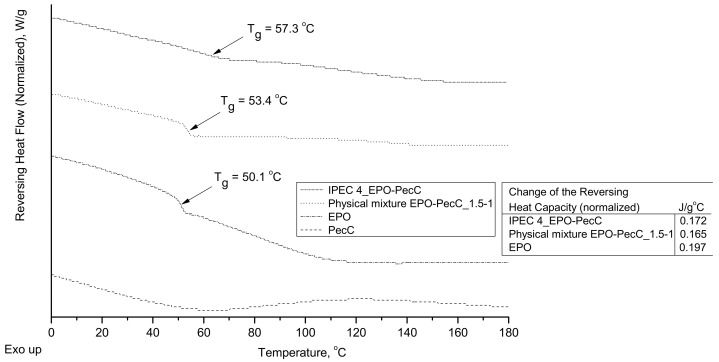
Results of DSC analysis of the IPEC EPO/PecC_4 sample, physical mixture IPEC EPO/PecC_1.5:1, and individual polymers.

**Figure 6 ijms-24-17622-f006:**
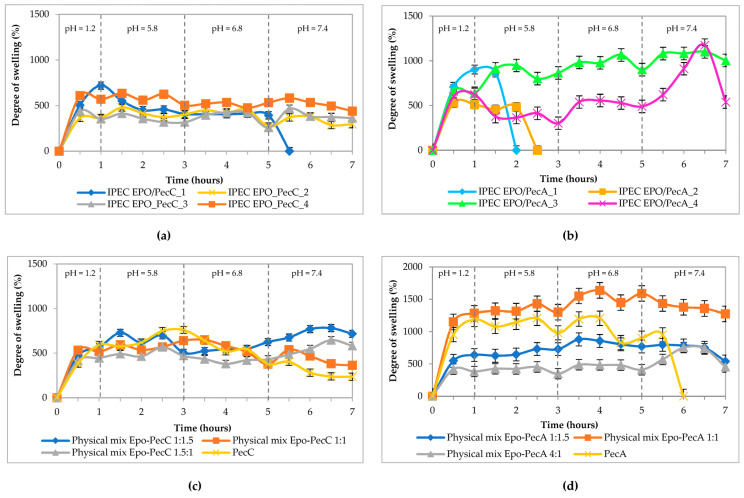
Kinetics of swelling of the studied samples. (**a**,**b**) IPEC samples; (**c**,**d**) physical mixtures and pectin.

**Figure 7 ijms-24-17622-f007:**
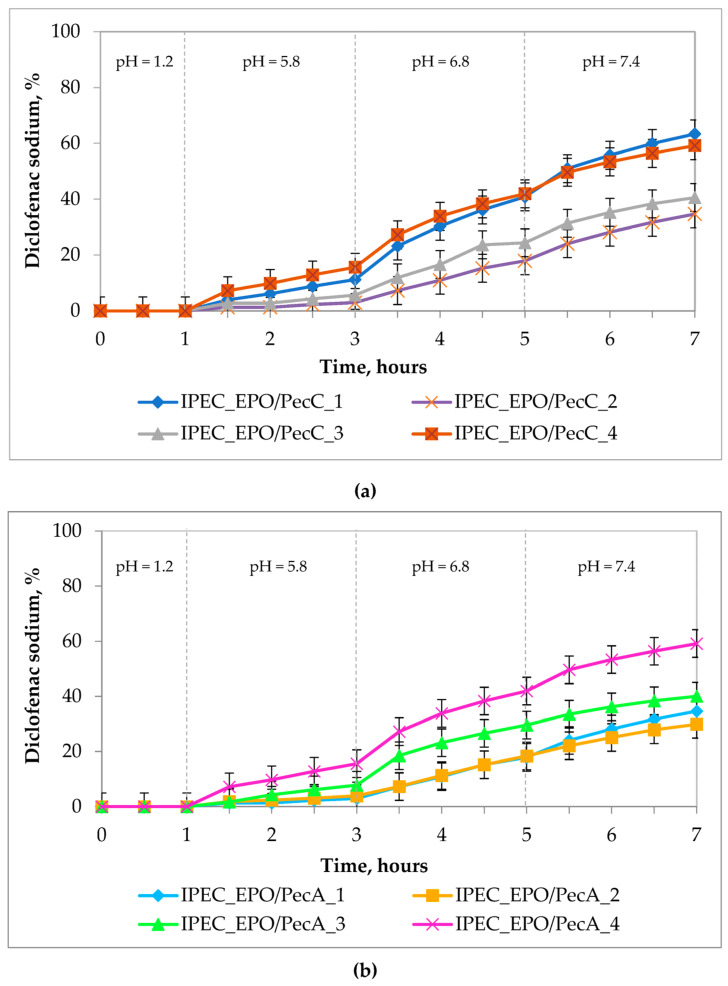
Kinetics of drug release of a model drug substance from IPEC matrices: (**a**) based on EPO/PecC; (**b**) based on EPO/PecA.

**Figure 8 ijms-24-17622-f008:**
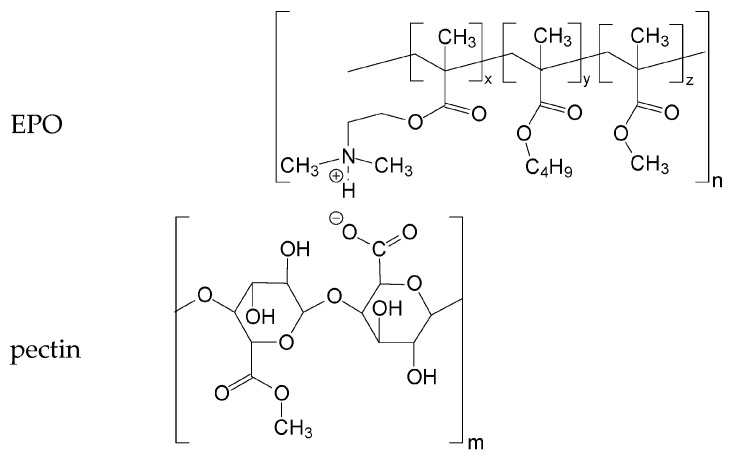
Scheme of interaction between EPO and pectin.

**Table 1 ijms-24-17622-t001:** Elemental results.

pH	EPO/PecC (mole/mole)	EPO/PecA (mole/mole)
4.0	1:1.74	1:1.8
5.0	1:1.41	1:1.67
6.0	1:1.35	1:1.38
7.0	1.4:1	1.78:1

**Table 2 ijms-24-17622-t002:** Sample symbols.

Sample Symbol	Molar Ration of Polymers EPO/Pec	pH at Which IPEC Was Obtained
IPEC EPO/PecC_1	1:1.5	4.0
IPEC EPO/PecC_2	1:1	5.0
IPEC EPO/PecC_3	1:1	6.0
IPEC EPO/PecC_4	1.5:1	7.0
IPEC EPO/PecA_1	1:1.5	4.0
IPEC EPO/PecA_2	1:1.5	5.0
IPEC EPO/PecA_3	1:1	6.0
IPEC EPO/PecA_4	4:1	7.0

**Table 3 ijms-24-17622-t003:** Results of mathematical modeling of drug release from IPEC matrices according to the Korsmeyer–Peppas and logistic equations.

The Korsmeyer–Peppas Equation		Mt/M_∞_ = k·t^n^ y = a·x^b^		
Parameters	IPEC_EPO/PecC_1	IPEC_EPO/PecC_2	IPEC_EPO/PecC_3	IPEC_EPO/PecC_4
Exponential release (*n*)	14.0 ± 1.8	4.8 ± 0.8	8.4 ± 0.9	16.4 ± 1.6
Constant release (*k*)	0.8 ± 0.1	1.0 ± 0.1	0.8 ± 0.1	0.7 ± 0.1
Correlation coefficient (*R*^2^)	0.938	0.957	0.963	0.945
Transport mechanism	Super Case II	Super Case II	Super Case II	Super Case II
	IPEC_EPO/PecA_1	IPEC_EPO/PecA_2	IPEC_EPO/PecA_3	IPEC_EPO/PecA_4
Exponential release (*n*)	0.5 ± 0.1	0.8 ± 0.2	2.6 ± 0.6	5.1 ± 0.8
Constant release (*k*)	2.2 ± 0.1	1.9 ± 0.1	1.5 ± 0.1	1.3 ± 0.1
Correlation coefficient (*R*^2^)	0.985	0.981	0.958	0.973
Transport mechanism	Anomalous transport	Anomalous transport	Super Case II	Super Case II
Logistic equation	y = A2 + (A1 − A2)/(1 + x/x0)˄p			
	IPEC_EPO/PecC_1	IPEC_EPO/PecC_2	IPEC_EPO/PecC_3	IPEC_EPO/PecC_4
A1	0.6 ± 0.4	1.5 ± 1.0	3.5 ± 0.9	6.4 ± 1.0
A2	56.1 ± 1.0	63.6 ± 2.5	83.2 ± 1.3	76.2 ± 1.3
X0	5.1 ± 0.1	4.7 ± 0.2	4.0 ± 0.1	3.7 ± 0.1
P	2.8 ± 0.1	2.1 ± 0.2	2.7 ± 0.1	2.4 ± 0.1
Correlation coefficient (*R*^2^)	0.998	0.996	0.998	0.997
	IPEC_EPO/PecA_1	IPEC_EPO/PecA_2	IPEC_EPO/PecA_3	IPEC_EPO/PecA_4
A1	1.0 ± 0.4	1.7 ± 0.2	1.4 ± 0.8	6.6 ± 0.1
A2	46.5 ± 3.7	35.9 ± 1.3	43.2 ± 2.5	67.7 ± 4.2
X0	5.5 ± 0.2	5.0 ± 0.1	4.0 ± 0.2	4.4 ± 0.2
P	4.3 ± 0.4	4.5 ± 0.3	4.2 ± 0.5	3.8 ± 0.4
Correlation coefficient (*R*^2^)	0.997	0.998	0.993	0.995

**Table 4 ijms-24-17622-t004:** Order and mixing ratios of polymers for turbidity measurements.

Mixing Order	Polymer Ratio								
EPO/PecC(or PecA)	9:1	8:2	7:3	6:4	5:5	4:6	3:7	2:8	1:9
PecC(or PecA)/EPO	9:1	8:2	7:3	6:4	5:5	4:6	3:7	2:8	1:9

**Table 5 ijms-24-17622-t005:** Molar ratios of polymers for viscosity measurements.

Molar Ratios EPO/PecC(or PecA)												
6:1	5:1	4:1	3:1	2:1	1.5:1	1:1	1:1.5	1:2	1:3	1:4	1:5	1:6

## Data Availability

Data are contained within the article.

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
