# Peer review of "Characteristics of Interpolyelectrolyte Complexes Based on Different Types of Pectin with Eudragit® EPO as Novel Carriers for Colon-Specific Drug Delivery"

_ijms, 2023, doi:10.3390/ijms242417622_

Round 1

Reviewer 1 Report

Comments and Suggestions for Authors

The Authors reported some formulations based on citrus and apple pectin and Eudragit® EPO used as carriers for oral drug delivery.

The authors need to address the following points before it can be accepted for publication:

In the introduction section, the Authors should point out better the aim and the novelty of the work. Moreover, I suggest to increase the number of references regarding the use of polysaccharide based gels for drug delivery. Among others, I recommend to see and cite these papers:

-         -  ACS Biomater. Sci. Eng. 2021, 7, 9, 4102–4127. DOI: 10.1021/acsbiomaterials.0c01784

-        -   Soft Matter. 2022. 18, 6443. DOI: 10.1039/d2sm00889k

Materials and methods section:  the Authors need to better describe the experimental procedure. In paragraph 4.2.1, in which solvent the polymers were solubilized? Water? Please specify. In 4.2.2 paragraph, modify table 5 because it is not clear. Moreover, revise lines 328 and 331. In 4.2.4 paragraph, specify and indicate the quantity of polymers used, the amount of acetic acid, NaOH and pectin solutions added.

The conclusion paragraph should be rewritten underlining the innovative contribution of the work in light of the found results.

Comments on the Quality of English Language

The whole manuscript requires a English revision because some sections are hard to read and, sometime, it becomes quite difficult to understand the actual message from the Authors

Author Response

Dear reviewer, thank you for the work done on reviewing our manuscript. We tried to answer questions and make adjustments in the manuscript.

Point 1:  In the introduction section, the Authors should point out better the aim and the novelty of the work. Moreover, I suggest to increase the number of references regarding the use of polysaccharide based gels for drug delivery. Among others, I recommend to see and cite these papers:

-         -  ACS Biomater. Sci. Eng. 2021, 7, 9, 4102–4127. DOI: 10.1021/acsbiomaterials.0c01784

-        -   Soft Matter. 2022. 18, 6443. DOI: 10.1039/d2sm00889k

      Response 1: Changes in the formulation of the purpose and novelty of the work have been made to the text. Links and references to these sources are included in the text and in the bibliography.

Point 2:  Materials and methods section:  the Authors need to better describe the experimental procedure. In paragraph 4.2.1, in which solvent the polymers were solubilized? Water? Please specify. In 4.2.2 paragraph, modify table 5 because it is not clear. Moreover, revise lines 328 and 331. In 4.2.4 paragraph, specify and indicate the quantity of polymers used, the amount of acetic acid, NaOH and pectin solutions added.

Response 2:

Synthesis example: IPEC ЕРО/РесС рН = 7.0

polymer ratio 1.5 : 1

synthesis for 2 L: m (PecC) = 0.0882 g, m (EPO) = 2.085 g

РесС solution preparing: 500 ml 0.05 М NaOH solution was added to powder of PecC, mixed on magnit stirrer up to dissolving.   0.05 М CH3COOH was added up to рН = 7.0, next solution added (mixture of 0.05 М NaOH+0.05 М CH3COOH рН = 7.0) up to 1L.

ЕРО solution: 400ml 0.05 М CH3COOH was added to EPO powder, mixed on magnit stirrer up to dissolving. 0.05 М NaOH was added slowly up to рН = 7.0 in previous solution, next solution added (mixture of 0.05 М NaOH+0.05 М CH3COOH рН = 7.0) up to 1L.

A PecC solution is added to the EPO solution (dropwise, approximately 150-180 drops/min). After 24 hours, the supernatant liquid is drained and the precipitate is washed 3 times with distilled water.

The remaining complexes were prepared similarly. Only the mass of polymers and the initial volumes of acid or alkali change.

Mole of EPO : Mole of PecС (or PecA)

6:1

5:1

4:1

3:1

2:1

1.5:1

1:1

1:1.5

1:2

1:3

1:4

1:5

1:6

Point 3:  The conclusion paragraph should be rewritten underlining the innovative contribution of the work in light of the found results

Response 3: The conclusion has been supplemented.

Reviewer 2 Report

Comments and Suggestions for Authors

The paper is focused on the characterization of polymeric carriers for the diclofenac release under variable pH conditions. The topic falls within the scope of the journal. The presentation and discussion of the results could be improved. On this basis, I recommend the publication after the following revisions:

-          Lines 117-120. The authors stated “Based on the results of turbidimetry, graphs representing a typical turbidimetric titration curve were constructed, which have a maximum at certain ratios of polymers; when an excess amount of pectin or EPO macromolecules is added, turbidimetry decreases and IPECs precipitates”. Why an excees of one polymer induces the sedimentation of IPECs? Details should be provided and related references with similar results should be quoted.

-          The quality of Figurse 3,4 are very poor. Please improve it.

-          DSC. I suggest to calculate the heat capacity change related to the glass transition process from the analysis of DSC curves.

-          Release data. I suggest to include the fitting curve obtained by the Krosmeyer-Peppas equation within the Figure 7 to have a direct comparison between experimental between experimental and fitted data.

-          Please report the Krosmeyer-Peppas equation.

-          Release data evidenced an induction time. The Krosmeyer-Peppas equation does not consider this parameter. Is it possible to use another fitting model that takes into account this parameter?

-          The fitting parameters obtained from Krosmeyer-Peppas equation could be compared with literature results related to drug release from other biopolymers, such as Funori, alginate and chitosan [International Journal of Biological Macromolecules, 234, 123645, 2023; New J. Chem., 2019,43, 10887-10893].  

Comments on the Quality of English Language

Minor corrections are needed. 

Author Response

Dear reviewer, thank you for the work done on reviewing our manuscript. We tried to answer questions and make adjustments in the manuscript.

Point 1: Lines 117-120. The authors stated “Based on the results of turbidimetry, graphs representing a typical turbidimetric titration curve were constructed, which have a maximum at certain ratios of polymers; when an excess amount of pectin or EPO macromolecules is added, turbidimetry decreases and IPECs precipitates”. Why an excees of one polymer induces the sedimentation of IPECs? Details should be provided and related references with similar results should be quoted.

Response 1: During turbidimetric titration, interaction between two polyelectrolytes occurs, which leads to an increase in turbidity in the solution as the concentration of polymers increases to a certain maximum. The maximum turbidity value corresponds to the equimolar ratio of PE and the formation of stoichiometric IPEC. Further addition of PE leads to a decrease in turbidity due to the precipitation of sediment.

Point 2: The quality of Figures 3,4 are very poor. Please improve it.

Response 2: We have done IR-spectra in the same program (Origin).

Point 3: DSC. I suggest to calculate the heat capacity change related to the glass transition process from the analysis of DSC curves.

Response 3: We have added additional information about the reversing heat capacity change in Fig. 5.

Point 4: Release data. I suggest to include the fitting curve obtained by the Krosmeyer-Peppas equation within the Figure 7 to have a direct comparison between experimental between experimental and fitted data.

Response 4: we didn't include fitting curve within drug release Figures in manuscript, but add here and remake fitting obtained by Logistic equation, which more suitable for our results. For compare we attach fitting curves of one IPEC based on apple and citrus pectin obtained by two equations. All samples results calculated and presents in manuscript.

  1. The fitting curve obtained by the Krosmeyer-Peppas equation (IPEC_EPO/PecC_1)

  1. The fitting curve obtained by the Logistic equation (IPEC_EPO/PecC_1)
  2. The fitting curve obtained by the Krosmeyer-Peppas equation (IPEC_EPO/PecA_1)
  3. The fitting curve obtained by the Logistic equation (IPEC_EPO/PecA_1)

Point 5: Please report the Krosmeyer-Peppas equation.

Response 5: We added equation in table.

Point 6: Release data evidenced an induction time. The Krosmeyer-Peppas equation does not consider this parameter. Is it possible to use another fitting model that takes into account this parameter?

Response 6: We took your opinion into account and recalculated the fitting using a different model – Logistic function.

Point 7: The fitting parameters obtained from Krosmeyer-Peppas equation could be compared with literature results related to drug release from other biopolymers, such as Funori, alginate and chitosan [International Journal of Biological Macromolecules, 234, 123645, 2023; New J. Chem., 2019,43, 10887-10893].  

Response 7:  We compared our fitting results with literature related to drug release from other biopolymers, fitting parameters were very different. So, we will try to use another model – Logistic. A Logistic function is describing the exponential rise of concentration and it’s the levelling off as growing experimental time. This function more suitable for our system and have better correlation coefficients.

Reviewer 3 Report

Comments and Suggestions for Authors

The reviewed article addresses an important topic for current pharmaceutical research. The authors propose optimal combinations of interpolar electrolytes and go through a rich, complete experimental program to test the final forms obtained. However, from the analysis of the material, there are important observations and recommendations to be made for its correction and improvement, in accordance with the publication requirements and high quality of the content:

1. A total restructuring of the Introduction chapter is recommended: thus, are definitions of acronyms that do not appear in the flow of the text when these acronyms appear for the first time (for example IPEC - the explicitness appears towards the end of the Introduction section, although the first time appears at the beginning).

- the ordering of all the information in this section must be rethought, for a fluent and intelligible flow (for example, reference is made to Eudragit, in combinations, then the substance is presented as such, and with its source!!); the city of a group of authors/researchers is done with a supplement (example at Moustafine  &col.).

- Presentation of the final purpose of the study must be done in more detail, and not as the little last part of a statement at the end of the Introduction Section.

1a. Attention to the ethical and promotional aspects of some products, respectively to advertising aspects (Eudragit:” Eudragit", produced by the company "Evonik Ind.", Germany).

2. Maximum attention to the ordering of the sections (Results, Discussions, Materials and Methods... in strict accordance with the requirements of the Manuscript Drafting Guide).

3. In the Materials and Method chapter, the experimental devices used are not sufficiently described, for example for the failure kinetics experiments, the type of device...the method as such used...).

4. The resolution of the figures must be uniform.

5. For the Conclusions chapter, the authors are asked to specify the openings for future studies, starting from their own results with more accuracy.

6. The recommendation would be that new and more recent ones should be included in the bibliographic references.

Comments on the Quality of English Language

Extensive editing of English language required

Author Response

Dear reviewer, thank you for the work done on reviewing our manuscript. We tried to answer questions and make adjustments in the manuscript.

The reviewed article addresses an important topic for current pharmaceutical research. The authors propose optimal combinations of interpolar electrolytes and go through a rich, complete experimental program to test the final forms obtained. However, from the analysis of the material, there are important observations and recommendations to be made for its correction and improvement, in accordance with the publication requirements and high quality of the content:

Point 1: A total restructuring of the Introduction chapter is recommended: thus, are definitions of acronyms that do not appear in the flow of the text when these acronyms appear for the first time (for example IPEC - the explicitness appears towards the end of the Introduction section, although the first time appears at the beginning).

Response1: It was corrected. The manuscript was rewritten.

Point 2: the ordering of all the information in this section must be rethought, for a fluent and intelligible flow (for example, reference is made to Eudragit, in combinations, then the substance is presented as such, and with its source!!); the city of a group of authors/researchers is done with a supplement (example at Moustafine &col.).

Response 2: The manuscript was reorganised and re-optimised. Additional information is included to the text of the manuscript.

Point 3: Presentation of the final purpose of the study must be done in more detail, and not as the little last part of a statement at the end of the Introduction Section.

Response 3: Introductory part is modified. Some additional information is included.

Point 4: 1a. Attention to the ethical and promotional aspects of some products, respectively to advertising aspects (Eudragit:” Eudragit", produced by the company "Evonik Ind.", Germany).

Response 4: Of course, we are grateful to the German company “Evonik Ind” for providing sample of Eudragit® copolymer. If the reviewer is thinking that this information is somehow contain the ethical and promotional aspects of their product (Eudragit) we suggest to move it to the Acknowledgment section of the manuscript.

Point 5: Maximum attention to the ordering of the sections (Results, Discussions, Materials and Methods... in strict accordance with the requirements of the Manuscript Drafting Guide).

Response 5: It was corrected.

Point 6: In the Materials and Method chapter, the experimental devices used are not sufficiently described, for example for the failure kinetics experiments, the type of device...the method as such used...).

Response 6: The manuscript was reorganised and re-optimised. Additional information is included to the text of the manuscript.

Point 7: The resolution of the figures must be uniform.

Response 7: We have increased the resolution of the figures by redraw them in a special program (Origin) and include them into the manuscript.

Point 8: For the Conclusions chapter, the authors are asked to specify the openings for future studies, starting from their own results with more accuracy.

Response 8: Additional information is included to the text of the Conclusion section of the manuscript.

Point 9: The recommendation would be that new and more recent ones should be included in the bibliographic references.

Response 9: Additional references were included to the bibliographic references.

Point 10: Extensive editing of English language required.

Response 10: The manuscript was carefully proofread and corrected.

Reviewer 4 Report

Comments and Suggestions for Authors

The authors report a study on the properties of different carriers for drug delivery based on pectin and Eudragit® EPO. Although the reported issues may have relevance in drug delivery, I believe that this manuscript can’t be accepted for publication in this Journal since it mainly lacks in novelty. The same research group has already reported a study on very similar systems as potential carriers for oral controlled drug delivery to the colon (see A.V. Bukhovets et al., Polymers 2020, 12(7), 1459; https://doi.org/10.3390/polym12071459). Furthermore, the colon-specific delivery should be investigated by in vitro/in vivo biological studies; in the absence of these studies, it makes no sense to discuss a specific release in the colon. 

Comments on the Quality of English Language

Moderate editing of English language required.

Author Response

Dear reviewer, thank you for the work done on reviewing our manuscript. We tried to answer questions and make adjustments in the manuscript.

Point 1 The authors report a study on the properties of different carriers for drug delivery based on pectin and Eudragit® EPO. Although the reported issues may have relevance in drug delivery, I believe that this manuscript can’t be accepted for publication in this Journal since it mainly lacks in novelty. The same research group has already reported a study on very similar systems as potential carriers for oral controlled drug delivery to the colon (see A.V. Bukhovets et al., Polymers 2020, 12(7), 1459; https://doi.org/10.3390/polym12071459). Furthermore, the colon-specific delivery should be investigated by in vitro/in vivo biological studies; in the absence of these studies, it makes no sense to discuss a specific release in the colon.

Response 1
Thank you for this comment. Regarding to our previously published paper A.V. Bukhovets et al., Polymers 2020, 12(7), 1459; https://doi.org/10.3390/polym12071459) where we have studied thecomplexation between Eudragit® EPO (polybase and Eudragit ® S 100 (poly acid) in various organic solvents (ethanol, tetrahydrofuran and isopropanol-acetone mixture). Of course, using of organic solvents can significantly simplify the synthesis of Interpolymer complexes (IPCs - stabilized preferably by hydrogenand hydrophobic bonds) compared to aqueous media, since there is no need to control the pH of the solutions. But for Interpolyelectrolyte complexes (IPECs-stabilized by ionic bonds) made up with participation of polysaccharides (in our case pectin) this method by using organic solvent media for preparing them is not useful. So, suggested combination of polymers apple pectin and citrus pectin with Eudragit® EPO was used for the first time to produce IPECs. In our work, new delivery systems are potentially suitable for colon specific drug delivery ; since they show a characteristic lag phase in the first hours of release in stomach and small intestine mimicking fluids , followed by an increase in the amount of DS release after moving to the large intestinal mimicking conditions with pH of 6.8 and 7.4 values.

We think that regarding to the reviewer suggestions we definitely need to add the word “potentially” claiming that our IPEC could be used as carriers for colon specific drug delivery system. This will help to avoid overemphasizing this claim.

Thank you for this suggestion. We agree that the in vitro/in vivo biological studies will be of interest for further optimisation of developed dosage forms. However, we believe it is outside of the scope of this manuscript. We may perform this study in the future and report it separately.

Point 2 Moderate editing of English language required.

Response 2 The manuscript was carefully proofread and corrected.

Reviewer 5 Report

Comments and Suggestions for Authors

Comments:

1.     In line 3, the title, there was a gap in E PO, please correct.

2.     The whole manuscript all the numbers like Figure 1, ph 4,0 5,0 6,0, Table 1, 1: 1,74; 1:1,8, Figure 7. Those have been corrected to 4.0, 5.0, 6.0 to make it readable.

3.     What are the size, PDI, zeta potential of the synthesized complex? Those are important data and should be provided.

4.     Figure 2, are those values statistical differences? They also should provide statistical analysis of those data.

5.     The FTIR data in Figure 3 and Figure 4, one with the grid, and one without. The authors should make this consistent.

6.     Figure 7b, the author changed the pH from 1.2 to 7.4 and monitored the release profile. Why did the author chose that way to monitor the release profile? Why don’t they monitor the release profile in pH 1.2, 5.8, 6.8 and 7.4 separately, and each pH they measure for 7h.

7.     In L 302, the authors mentioned the systems are suitable for colon-specific drug delivery. What is the safety and the bioactivity of the materials? The authors should provide those data, then can say it is suitable for in vivo delivery, only in vitro release profile is too thin.

8.     What is the novelty of using this pectin with EPO carriers compared to other drug carriers for colon-specific drug delivery?

Comments on the Quality of English Language

Lots of formatting errors need to correct. 

Author Response

Dear reviewer, thank you for the work done on reviewing our manuscript. We tried to answer questions and make adjustments in the manuscript.

Point 1: In line 3, the title, there was a gap in E PO, please correct.

Response 1:  It was corrected.

Point 2: The whole manuscript all the numbers like Figure 1, ph 4,0 5,0 6,0, Table 1, 1: 1,74; 1:1,8, Figure 7. Those have been corrected to 4.0, 5.0, 6.0 to make it readable.

Response 2:  It was corrected.

Point 3: What are the size, PDI, zeta potential of the synthesized complex? Those are important data and should be provided.

Response 3: We did not study the size and charge of particles due to the fact that we did not developing nano- or micro-sized particles, due to, in frame of this manuscript, we’ve prepared a matrix-based systems (tableting matrices) by using synthesized IPECs.

Point 4: Figure 2, are those values statistical differences? They also should provide statistical analysis of those data.

Response 4: We have added the results of statistical analysis to the manuscript.

Point 5: The FTIR data in Figure 3 and Figure 4, one with the grid, and one without. The authors should make this consistent.

Response 5: We have done IR-spectra in the same program (Origin) and include them into the manuscript.

Point 6: Figure 7b, the author changed the pH from 1.2 to 7.4 and monitored the release profile. Why did the author choose that way to monitor the release profile? Why don’t they monitor the release profile in pH 1.2, 5.8, 6.8 and 7.4 separately, and each pH they measure for 7h.

Response 6: We used sequential change of pH during drug delivery experiment because want to mimicking drug transit trough gastrointestinal tract. Our systems include polymers that are pH- and time- dependent, the IPEC matrices swell as they move through the different parts of GI tract, thus drug delivery occurs because the matrices swell as pH changes and as matrix residence time increases.

Point 7: In L 302, the authors mentioned the systems are suitable for colon-specific drug delivery. What is the safety and the bioactivity of the materials? The authors should provide those data, then can say it is suitable for in vivo delivery, only in vitro release profile is too thin.

Response 7: Eudragit EPO polymer has been used in the pharmaceutical industry for many decades as excipient in developing oral dosage forms, and therefore it can be concluded that this polymer is safe (Eisele, J., Haynes, G. Rosamilia, T. Characterization and toxicological behavior of Basic Methacrylate Copolymer for GRAS evaluation. Regulatory toxicology and Pharmacology, 2011, Volume 61, Issue 1p. 32-43. DOI:10.1016/j.yrtph.2011.05.012)

Pectin is a biodegradable polysaccharide of natural origin, obtained from apples and citrus fruits, which is desitegrated as it is digested in the intestines. Thus, IPECs obtained on the basis of studied polymers are safe for use as delivery systems. IPECs based on pectin and Eudragit EPO were tested for toxicity. However, the results are not presented in this work, since this was not the aim of this work.

Toxicity results: The resulting systems were tested for toxicity. The study used 48 white male mice weighing 20-25 g and 4 months old. Aqueous solutions of the studied compounds were administered intragastrically once using a special probe. Toxicity was studied at doses of 1000 mg/kg and 2000 mg/kg, each administered intragastrically to 6 mice. The control group was administered distilled water in a similar manner. Throughout the entire observation period, the appearance, behavior of animals and the dynamics of body weight gain in the experimental group did not differ from those of mice in the control group. There was no decrease in food consumption, no signs of intoxication or death of mice in both groups.

Thus, analysis of the results of studying the acute toxicity of compounds after a single intragastric administration at doses of 1000 mg/kg and 2000 mg/kg did not reveal intoxication and death of animals.

In addition, release was assessed in the presence of pectinase to approximate intestinal conditions (Figure below). No statistical difference was observed between the results, so we continued the experiments without pectinase.

Point 8: What is the novelty of using this pectin with EPO carriers compared to other drug carriers for colon-specific drug delivery?

Response 8: This combination of polymers pectin with EPO was used for the first time to produce IPECs. In our work, new delivery systems were obtained based on apple pectin and citrus pectin with Eulragit® EPO, it was proven that these are chemically individual compounds, and the prospects of using the obtained IPECs for modified drug delivery to the intestine (colon) region were shown.

Round 2

Reviewer 1 Report

Comments and Suggestions for Authors

I accept the manuscript in present form

Author Response

Thank you very much for your positive decision and thanks again for evaluation and very useful recommendations which significantly improved our manuscript.

Reviewer 2 Report

Comments and Suggestions for Authors

The paper was revised according to the reviewers' suggestions. In my opinion, the paper can be published after the following revision:

- release data. The physical meaning of the fitting parameters obtained from the logistic equation should be presented and discussed. A comparison of the parameters for the different systems should be conducted. 

Comments on the Quality of English Language

Minor corrections are needed.

Author Response

Response to Reviewer 2 Comments

Dear reviewer, thank you for the work done on reviewing our manuscript. We tried to answer questions and make adjustments in the manuscript.

Point 1: - release data. The physical meaning of the fitting parameters obtained from the logistic equation should be presented and discussed. A comparison of the parameters for the different systems should be conducted. 

Response 1: Some additional information has been added to the manuscript (Pages 15-16, lines 391-414).

Reviewer 3 Report

Comments and Suggestions for Authors

The manuscript has been sufficiently improved to warrant publication.

Comments on the Quality of English Language

The manuscript must be revised for minor English editing .

Author Response

Response to Reviewer 3 Comments

Dear reviewer, thank you for the work done on reviewing our manuscript. We tried to answer questions and make adjustments in the manuscript.

Point 1: The manuscript has been sufficiently improved to warrant publication. The manuscript must be revised for minor English editing 

Response 1: The manuscript was carefully proofread and corrected.

Reviewer 4 Report

Comments and Suggestions for Authors

I still believe that this manuscript can’t be accepted for publication in this Journal since it  lacks in novelty. Furthermore, the word “potentially” can't justify the use of these materials for colon specific drug delivery system. In vitro/in vivo biological studies are needed.

Comments on the Quality of English Language

Minor editing of English language required

Author Response

Response to Reviewer 4 Comments

Dear reviewer, thank you for the work done on reviewing our manuscript. We tried to answer questions and make adjustments in the manuscript.

Point 1: I still believe that this manuscript can’t be accepted for publication in this Journal since it lacks in novelty. Furthermore, the word “potentially” can't justify the use of these materials for colon specific drug delivery system. In vitro/in vivo biological studies are needed. 

Response 1: Thanks again for your opinion and given suggestion. Of course, we agree with you, however the carrying out of the in vivo experiments need much more time and efforts for evaluation of the results and including them. Thus, in our group we did the in vivo experiments later and published the results separately. For proving that, please see our previously published papers with in vivo data and evaluation of them:

  • Mustafin, R.I.; Kabanova, T.V.; Semina, I.I.; Bukhovets, A.V.; Garipova, V.R.; Shilovskaya, E.V.; Nasibullin, Sh. F.; Sitenkov, A.Yu.; Kazakova, R.R.; Kemenova, V.A. Biopharmaceutical assessment of polycomplex matrix system based on Carbomer 940 and Eudragit® EPO for colon-specific drug delivery. Chem. J., 2011, 45, 491-494. DOI: 0091-150X/11/4508-0491
  • Mustafin, R.I.; Semina, I.I.; Bukhovets, A.V., Sitenkov A.Yu., Garipova V.R., Salakhova A.R., Kemenova V.A. Comparative pharmacokinetic assessment of diclofenac sodium polycomplex drug delivery systems based on Eudragit copolymers. Pharm. Chem. J., 2014, 48, 1-4. DOI: 0091-150X/14/481-0001
  • Mustafin R.I., Semina I.I., Garipova V.R., Bukhovets A.V., Sitenkov A.Yu., Salakhova A.R., Gennari C.G.M., Cilurzo F. Comparative study of polycomplexes based on Carbopol® and oppositely charged polyelectrolytes as a new oral drug delivery system. Chem. J., 2015, 49, 1-6. DOI: 0091-150X/15/4901-0001
  • Timergalieva (Garipova), V.R.; Sitenkov, A.Yu.; Bukhovets (Sitenkova), A.V.; Elizarova, E.S.; Gordeeva D.S.; Semina, I.I.; Moustafine, R.I. Development of lyophilisates based on polymer-drug and interpolyelectrolyte complexes: pharmacokinetic assessment. Drug Dev. & Reg. 2023, 12, 173–18 DOI:10.33380/2305-2066-2023-12-4-159

According to the above-mentioned papers that we published, our assumptions about possibility for using polycomplex carriers based on Eudragits and oppositely charged polymers for the development of oral DDS, including colon-specific delivery systems, could be reasonable.   

Point 2: Minor editing of English language required.

Response 2: The manuscript was carefully proofread and corrected.

Reviewer 5 Report

Comments and Suggestions for Authors

All the comments are addressed. 

Author Response

(The authors gave the same response as above.)
